# Non-Standard Map Robot Path Planning Approach Based on Ant Colony Algorithms

**DOI:** 10.3390/s23177502

**Published:** 2023-08-29

**Authors:** Feng Li, Young-Chul Kim, Boyin Xu

**Affiliations:** 1Department of Mechanical Engineering, Kunsan National University, Gunsan 54150, Jeollabuk, Republic of Korea; fengli001@126.com (F.L.); xuboyin49@gmail.com (B.X.); 2College of Smart Manufacturing, Zhengzhou University of Economics and Business, Zhengzhou 450007, China

**Keywords:** robot path planning, non-standard map, calibration, grid map, ant colony algorithm

## Abstract

Robot path planning is an important component of ensuring the robots complete work tasks effectively. Nowadays, most maps used for robot path planning obtain relevant coordinate information through sensor measurement, establish a map model based on coordinate information, and then carry out path planning for the robot, which is time-consuming and labor-intensive. To solve this problem, a method of robot path planning based on ant colony algorithms after the standardized design of non-standard map grids such as photos was studied. This method combines the robot grid map modeling with image processing, bringing in calibration objects. By converting non-standard actual environment maps into standard grid maps, this method was made suitable for robot motion path planning on non-standard maps of different types and sizes. After obtaining the planned path and pose, the robot motion path planning map under the non-standard map was obtained by combining the planned path and pose with the non-standard real environment map. The experimental results showed that this method has a high adaptability to robot non-standard map motion planning, can realize robot path planning under non-standard real environment maps, and can make the obtained robot motion path display more intuitive and convenient.

## 1. Introduction

With the development of the times, robots have become indispensable assistants in various industries, among which mobile robots are more widely used due to their unique structure. At bus stops, they can replace humans in automatic response. In warehouses, they can replace humans for material handling. On the battlefield, they can even replace humans in charge, assault, etc. [1,2,3,4]. The unique functions of robots make them increasingly at the forefront of technological development and an indispensable tool for humanity.

The determination of the robot’s movement path is a crucial step in completing work tasks. The planning of robot movement path mainly refers to the route that the robot needs to pass through during the process of reaching another position from one location. This route not only ensures that the robot can reach the endpoint from the starting point but also ensures that the robot can avoid obstacles and dangerous objects on the path. With the development of technology, the intelligence level of robot movement path planning has increasingly become a focus of research for researchers. How robots can quickly and intelligently determine their movement paths can greatly optimize the autonomy and speed of mobile robot navigation. Therefore, the study of robot path planning method is of great significance [5].

In order to determine the robot’s motion path, it is first necessary to obtain the robot’s motion environment information, which is the robot’s motion environment map. Currently, robot mobile maps mainly include Occupancy Grid Map, Octo Map, and Point cloud Map. Due to the discreteness of data in computer, we need to perform a discreteness design on the x and y axes. In this way, we can obtain a large number of grids through discretization design on the x and y axes. Each grid can be represented by 0 or 1. This can generate a structured Occupancy Grid Map. Occupancy Grid Maps are typically generated through laser measurement of relevant data. It has a simple structure and is easy to implement various intelligent path planning algorithms for robots [6,7].

In addition, we can continuously split the 3D grid map into smaller cubes, label obstacles, and hazards. This forms an Octo map. An Octo map is a tree with eight subordinate nodes. It is often used for the expression of three-dimensional data [8,9]. We can divide the cube into eight small cubes and continue dividing the cubes containing obstacles or hazards until the requirements are met. When all the subordinate nodes of a node are occupied or idle or unknown, we can remove it and all its subordinate nodes. In addition, there is also a point cloud map that uses LiDAR to obtain the three-dimensional coordinates of different points and generate maps based on laser reflection intensity. It is precisely with the existence of these maps that the path planning for robots becomes possible. 

After the robot’s movement map is generated, various algorithms are also used to determine the robot’s movement path and pose. In 1956, Edsger Wybe Dijkstra proposed the Dijkstra algorithm. This algorithm centers around the initial point and gradually expands outward until the goal is achieved. The A* algorithm appeared in 1968 and optimized the Dijkstra algorithm through heuristic functions to find the optimal path faster [10]. In 1985, Khatib proposed an artificial potential field method, which assumes that obstacles have a repulsive force on the robot and the target object has gravitational force on the robot. The repulsive force and gravitational force are combined to form the motion path of the robot [11,12]. The random road map (PRM) algorithm is an algorithm that converts the continuous space state to the discrete space state and then uses search algorithms such as A* to find the path [13,14]. The rapidly exploring random tree (RRT) algorithm detects the collision of the acquisition points to avoid spatial modeling [15,16,17]. It effectively solves path planning problems in high-dimensional spaces and complex constraints. In addition, there is the genetic algorithm (GA) proposed by John Holland, which simulates the search for optimal solutions in natural evolutionary processes [18,19]. It is a computational model based on natural selection and genetics [20,21]. 

By combining various path planning algorithms with environmental maps, we can achieve path planning for mobile robots. Since ant colony algorithms have highly efficient searchability [22,23,24,25,26,27], it has also become a commonly used path planning method. This paper proposed a method to transform the non-standard real environment map into the standard robot mobile environment map-occupancy grid map, and to then use ant colony optimization algorithms to realize the robot moving path, position, and pose planning.

In this paper, a new method of ant colony algorithm planning robot motion path using a real environment map was proposed. This method processes non-standard real environment maps and converts them into standard occupancy grid maps. Through ant colony algorithms, the planning path and pose of the robot are given. Then, the robot’s planned path and pose are combined with the original non-standard map and generate a mobile path planning map for the robot in a non standard environment. The planning results of the virtual environment and the results of the real experimental environment have once again verified the application potential of this method.

The second part of the article discusses the origin of the research and introduces the practical research issues and related preparations. Part three introduces the overall solution approach, explains the method of converting detailed non-standard maps into standard maps, and the path planning was implemented in steps. The fourth part validated the method of path planning for robots under non-standard maps in real environments through experiments and conducted comparative analysis. The fifth part summarizes the article [28].

## 2. Problem Description

### 2.1. Realistic Issues

The planning of robot motion trajectory needs to be based on a map. Robot motion maps are mostly based on sensors such as infrared to explore surrounding objects, collect relevant information, and create maps based on the collected information, as shown in Table 1 [11,12,13,14,15,16,17,18,19,20,21,22,23,24,25,26,27,28,29,30,31,32,33,34,35,36,37]. The robot avoids different obstacles based on the formed map and gradually approaches the target through the planned motion path and pose requirements. However, information collection requires a lot of time, and the data processing and map generation after collection also require a lot of time. If the environmental information that needs to be collected is relatively complex, the time for establishing this map is quite long. Efficiently generating robot motion environment maps has become an urgent problem to be solved [38,39].

The development of artificial intelligence has made rapid progress. If we can obtain a real environment map by taking photos, process the real environment map, transform non-standard environment maps into robot path planning maps suitable for robot path planning, and then carry out path planning and pose planning for robots, this can greatly improve the efficiency of robot path planning. Grid maps are commonly used nowadays, and if non-standard maps can be converted into more common grid maps, subsequent processing will be more convenient. The purpose of this article is to process the non-standard real environment map to form a standard grid map, and then realize the robot’s motion path and pose planning through ant colony algorithms.

### 2.2. Processing of Non-Standard Maps

#### 2.2.1. Image Processing

The image-processing technology is relatively mature. It can be used for disease prediction [40], crop recognition [41], and industrial defective product selection [42]. We can use image-processing technology to segment and crop non-standard maps in real environments, locate, detect, and frame obstacles in the images, and identify obstacles. Subsequently, a new image is generated using the previously obtained image parameters, which is the standard robot motion path planning map that needs to be obtained. Because the image itself expresses the relative relationships between objects in the environment, using non-standard maps for image processing to obtain standard grid maps is more accurate and convenient.

#### 2.2.2. Obtaining Maps Adapted to Algorithms

The grid method is a commonly used map modeling method in robot path planning, which is simple and convenient to use. This article obtains relevant modeling parameters through non-standard real environment maps and uses the grid method to establish a map of the mobile robot’s mobile environment. In the map, 0 represents the area of the free area, and 1 represents the area of obstacles or dangerous objects in the map. The grid with a 0 value is represented in white, while the grid with a 1 value is represented in black (see Figure 1). This is similar to the principle of image binarization in image processing, but due to the fact that image processing is mostly based on pixels and the data are huge, we need to perform secondary processing after image processing. Alternatively, the non-standard map itself can be processed to obtain a standard map suitable for robot path planning.

#### 2.2.3. Method of Image Processing

After the image is read, the image is transformed into an array I. If the image is a real color array, the array I is an array of m × n × 3. Because the grid graph is a two-dimensional array of small squares, it is necessary to convert the image into a two-dimensional array. Therefore, it is necessary to perform binary processing on the image. We can convert the true color image into a grayscale image, and convert m × n × 3 arrays to m × n arrays. In this way, the non-standard map can be transformed into a standard grid map that can implement ant colony algorithms. The binarized image also needs to be subjected to obstacle recognition, positioning, and subsequent data processing, as detailed in Section 3.

### 2.3. Introduction to Ant Algorithm

#### 2.3.1. Basic Principles

Among intelligent algorithms, the ant colony algorithm is a common algorithm. This method mainly uses the method that each ant in the ant colony selects the moving route under the guidance of a pheromone to calculate. Because the accumulation degree of pheromone released by ants is different on different routes, ants will choose the route with more pheromone accumulation when moving. We can assume a directed graph *G*(*N*, *A*) with *N* locations, where *N* = {1, 2, 3, 4…, *n* − 1, *n*}, *A* = {(*i*, *j*)|*i*, *j* ∈ *N*}.

The scheduled path planning task is represented by the objective function:(1)$$min\;f\left(\omega \right)=\sum_{i=1}^{n}l_{i-1,i},$$
and ω = (*i*_1_, *i*_2_, *i*_3_, *i*_4_ⵈ *i_n_*_−1_, *i_n_*).

$l_{ij}$ is the distance between two points,
(2)$$l_{ij}=\sqrt{{\left(x_{i}-x_{j}\right)}^{2}+{\left(y_{i}-y_{j}\right)}^{2}},\;(i,\;j\;\in \;N)$$

When ants move from one position to the next, they can choose the next position based on the probability of reaching the next position.

This probability needs to be calculated using the following equation:(3)$$P_{ij}^{k}(t)=\frac{{\left[\tau_{ij}^{\left(t\right)}\right]}^{\alpha }\times {\left[\eta_{ij}^{\left(t\right)}\right]}^{\beta }}{\sum_{k\;\in \;allowed\;k}{\left[\tau_{ij}^{\left(t\right)}\right]}^{\alpha }\times {\left[\eta_{ij}^{\left(t\right)}\right]}^{\beta }},$$

Among them, *k* ∈ allowed *k*; *i*, *j* is the starting point and target point of the ants,
(4)$$\eta_{ij}=\frac{1}{l_{ij}},$$

$\eta_{ij}$ is visibility, which is the reciprocal of the distance between *i* and *j*; $\tau_{ij}^{\left(t\right)}$ is the pheromone strength from *i* to *j* at time *t*; allowed *k* is a collection of locations that have not been visited yet; *α*, *β* are two constants. They are the weighted values of pheromone and visibility.

In the process of ant colony algorithms, the pheromone in the algorithm is constantly updated.
(5)$$\tau_{ij}^{\left(t\right)}=(1-\rho )\tau_{ij}+\sum_{k=1}^{m}△\tau_{ij}^{k},$$

In the above formula, *m* is the number of ants in the ant colony. $△\tau_{ij}^{k}$ is the Pheromone left by the *k*-th ant when passing on the path from *i* to *j*.
(6)$$△\tau_{ij}^{k}={\left(C_{k}\right)}^{-1},$$

Among them, *i*, *j* are the points on the path passed by the ant. *C_k_* is the total length of the path obtained by the *k*-th ant walking along the entire path.

In this way, the ant completes the path once, iterates the operation once, and implements the algorithm through continuous iteration [43].

#### 2.3.2. Implementation

Usually, after generating a standard grid map, we can set the starting point and the target point to be reached in the planned path according to the requirements and set the basic parameters of the algorithm. The ant colony algorithm is used to search the path. The ant updates the pheromone on the path according to the algorithm after reaching the destination, outputs information, and saves the path information.

#### 2.3.3. Post-Processing

After the robot path planning is realized according to ant colony algorithms, it is also necessary to obtain the position and pose information of the robot movement according to the path information. After obtaining the path and pose information, it is necessary to combine the path and pose information with the non-standard real environment map to complete path planning and pose planning under non-standard maps. The entire process is a unified one. In this way, combined with image processing, ant colony algorithm, and pose planning algorithm, it is very convenient to realize robot path planning under non-standard maps [2,44,45].

#### 2.3.4. Related Work

There are many methods for robot path planning now. Jinghua Wang et al. combined Dijkstra’s algorithm and used a fuzzy logic system to consider the environmental conditions of path planning to study the path planning of mobile robots in complex two-dimensional terrain [46]. Oscar Loyola, John Kern, and Claudio Urrea proposed to generate the robot’s path by adding a boredom element without a reward [47]. Yigao Ning et al. proposed to add a safe distance to the existing repulsive potential energy field, and an obstacle avoidance controller was designed based on IAPF to study vehicle path planning under a multi-obstacle environment [48]. Novak Zagradjanin et al. proposed a cloud-based autonomous multi-robot system to perform highly repetitive tasks. The study further improved path planning in complex and crowded environments [49]. W Wei et al. studied the long-distance autonomous path planning of intelligent vehicles by using the Dijkstra algorithm and studied the autonomous obstacle avoidance of intelligent vehicles by using the artificial potential field method combined with the direction sensor and infrared sensor [50]. Hao Ge, Xin Li, Leiyan Yu et al. also studied the path planning method through different methods [51,52,53,54,55]. These methods implement and optimize the path planning from different aspects, but the establishment of the environment map is based on free modeling, and the path planning is not implemented in the real environment. Haichuan Zhang et al. proposed a search path design algorithm to improve the efficiency of maritime search and rescue. Based on the basic ant colony algorithm, they designed the search and rescue path of the ship at sea [56]. In this study, the designed maritime search and rescue path was finally represented on the navigation map. However, this study was still based on the existing parameters for modeling and generating maps for path planning and was not carried out in the real environment map.

It can be seen that the research of planning path algorithms on real environment maps is blank. However, applying the planning path algorithm to the real environment map can not only greatly improve the intelligence and practicability of the algorithm, but also greatly reduce the control difficulty of robots and vehicles. At present, car automatic driving is generally based on the combination of satellite navigation and sensors, if the path can be automatically planned under the real environment map, it will greatly im-prove the intelligence of automatic driving [57]. The path planning method under the real environment non-standard map of this article will be introduced in detail below.

This article first created a non-standard simulation map of mobile robot motion and used the non-standard simulation map to provide an overall introduction to the method proposed in the article. We also conducted experiments using the path planning method proposed in this article by displaying non-standard maps of the environment. The proposed method was experimentally validated. The path planning method for mobile robots under non-standard maps proposed in the article mainly included three steps.

Step 1: Process non-standard real environment maps to form a standard grid map for robot motion path planning;

Step 2: Plan the robot’s motion planning path according to the grid map;

Step 3: Integrate the path and pose planned by the grid map with non-standard real environment maps. Create a robot motion path and pose planning map under non-standard real environment maps.

## 3. Methods

Performing ACO on a real environment map required first normalizing the non-standard real environment map to generate a standard 2D grid map. Then, a 2D grid map and ant colony algorithm were used to plan the path from the initial location to the target location. At the same time, there were some obstacles in the non-standard map, and the planned path should not collide with these obstacles.

We read non-standard real environment maps using Matlab software. To standardize map processing, first, we selected the specified box selection area in the non-standard map, and we cut non-standard maps to obtain standard areas for establishing standard maps. Then, we performed binary processing on the cropped generated area. After the binarization of non-standard maps, obstacle recognition was performed on the map. In order to clearly reflect the actual dimensions on the map, there must be a scale calibration object inside the non-standard map. The dimensions of other objects and the planned path parameters should be referenced by the calibration object. This lay a good foundation for generating maps in non-standard environments that can reflect the real environment. Because the data after obstacle recognition were extracted by MATLAB, the binarized map was then subjected to two-dimensional grid processing. In order to facilitate the reading of positions on the map, it was necessary to encode the grid after grid processing. This made it very convenient to know the positions of various points on the map. After grid processing, non-standard maps were converted into standard two-dimensional grid maps. 

Next, path planning was carried out through the ant colony algorithm. After obtaining the planned path, we can proceed with pose planning based on the path parameters. After obtaining the path and pose, we read the non-standard map. Due to the use of size calibration objects in the early stage, the obtained path parameters and pose parameters can be converted into real environmental parameters. Based on these parameters, we can display the planned path and pose in the non-standard map, so as to obtain the planned path and pose of the non-standard map. The main process is shown in Figure 2.

### 3.1. Non-Standard Map Processing

For ease of explanation, we first established a virtual non-standard map, as shown in Figure 3. From the figure, it can be seen that each obstacle on the map had different sizes and irregular shapes. The circle in the upper right corner of Figure 3a represents the designated calibration object, and we assumed its diameter to be *L*. The size of other objects was based on it as a reference. In order to streamline, generalize, and standardize the entire process, we box-selected the non-standard map in Figure 3a with a size of *a* × *a*. Then, we obtained Figure 3b. After box selection, we cropped it to obtain Figure 3c. Next, we binarized Figure 3c to obtain Figure 3d,e, which are two different forms of image binarization. In order to facilitate subsequent processing and facilitate the identification of obstacles and calibration objects, we selected Figure 3d for subsequent processing.

### 3.2. Determination of Obstacles in Non-Standard Environmental Maps

After obtaining Figure 3d, we used the region props function provided by MATLAB to obtain the area parameters of the obstacle. We used the rectangle function for box selection, as shown in Figure 4. After the box selection, we not only obtained information about the obstacles but also obtained the size *L*_1_ of the calibration object in the figure. Though the size of the entire map was *a* × *a*, because the actual diameter of the calibration object was *L*, the actual size reflected in the entire diagram should be (*a* × *L*/*L*_1_) × (*a* × *L*/*L*_1_). From the figure, it can be seen that the boundary between the identified obstacle and the calibration object in the upper right corner was clearly visible. We placed the calibration object in the upper right corner to accurately size the map without affecting the robot’s final planning path.

### 3.3. Grid Map Design

We needed to segment the binary map into *n* × *n* grids to convert it into a grid map. This means that the map was divided into *n*^2^ grids. Because the size of the entire diagram was *a* × *a*, the side length of each grid was *u* = *a*/*n*. Next, we used the floor function to convert the matrix where the obstacle was located to 0, and the remaining areas to 1. Afterward, the fill function was used to fill the area with a median value of 0 in black, forming a standard grid map as shown in Figure 3f. In order to facilitate the subsequent ant colony algorithm path planning, we also needed to code each grid. Because the edge length of each grid was u, we divided the map into *n* × *n* parts, so the position of each encoding can be obtained in the following way. The encoding principle process is shown in Algorithm 1, and the encoded map is shown in Figure 5.
**Algorithm 1** Principle of location coding for non-standard environmental maps**Coding Principle**Input: Image edge length *a,* number of grids *n* × *n*Output: Grid edge length *u* = *a*/*n*1. Loop *n* times starting from the first line2. Loop *n* times starting from the first column3. The encoding position *y* for each loop row is *u* × *j*, and *j* is the current number of columns4. Each loop has a column encoding position *x* of *u* × *i*, where *i* represents the current number of rows5. The encoded value for each position is num2str ((*i* − 1) × *a*/*u* + *j*6. End column loop7. End row loop8. End

### 3.4. Algorithm Design

After obtaining the standard map, we used ant colony algorithms to plan the path. The specific process is shown in Figure 6. Due to the encoding of the map in the early stage, path planning for any two locations can be implemented by inputting the encoding. From Figure 5, it can be seen that we divided the image into 60 × 60 grids, which means there were a total of 3600 grids and 3600 positions in the entire image. Except for the areas marked as obstacles in the figure, other areas can be selected for path planning, which was convenient and simple.

In this virtual non-standard map, we set the starting position as point 130 and the target position as point 3513. The number of iterations here was 100. The number of ants was 100. These parameters can be increased or reduced, and the larger the value, the better the path planning effect. But the calculation time also increased. According to the previous research experience, we set the evaporation coefficient of the pheromone as 0.3, the increasing intensity coefficient of pheromone as 1, and the importance coefficient of the pheromone as 1 [28]. See the Table 2 for the other main parameters.

### 3.5. Path and Pose Generation

Figure 7 shows the planning path obtained through 100 iterations of the ant colony optimization algorithms. In the figure, grid 130 is the initial point of the design, and grid 3513 is the end point of the design. The blue route is the planned path. Ants pass through 57 points, as shown in Table 3, through points 130, 191, 252, 313, 374–3393, 3453, 3513, bypassing obstacles and reaching the target location.

Assume that the planning path is *R*, *R* = {*R*_1_, *R*_2_, …, *R*_i_, …, *R_n_*}, where *R*_1_ is the initial point and *Rn* is the target point. The coordinates of each point of *R_i_* is (*x_i_*, *y_i_*), then the point set coordinate formula:*C* = {(*x*_1_, *y*_1_), (*x*_2_, *y*_2_) …, (*x_i_*, *y_i_*), …, (*x_n_*, *y_n_*)};

The distance between each paragraph is *L_RiRi_*_+1_, *i* ∈ [1, *n* − 1],
(7)$$L_{Ri,Ri+1}=\sqrt{{\left(x_{Ri+1}-x_{Ri}\right)}^{2}+{\left(y_{Ri+1}-y_{Ri}\right)}^{2}}\;,\;I\in \;[1,\;n-1]$$

The total path length is *S*,
(8)$$S=\sum_{i=1}^{n-1}L_{Ri,Ri+1},$$

Then, the posture of each segment is *β_Ri,Ri+1_*,
(9)$$\beta_{Ri,Ri+1}=arctan\frac{y_{Ri+1}-y_{Ri}}{x_{Ri+1}-x_{Ri}},\;i\in \;[1,\;n-1].$$

Based on the above analysis, the planned displacement and position can be easily obtained. In order to find a good planning path, we chose the convergence method with a limited number of iterations. Figure 8 is the displacement convergence curve obtained by 100 iteration based on the above analysis. Figure 9 is the pose planning diagram obtained from the robot planning route in Figure 7.

### 3.6. Integration of Non-Standard Virtual Environment Maps and Path Planning

Because the virtual map path in Figure 7 was obtained from the original non-standard map, the obtained planning path and pose can be regressed to the original non-standard map. According to the data in Figure 7 and Figure 9, the path and position of ant optimization algorithms in the non-standard map can be obtained by displaying them in the original non-standard map. The thick blue line in Figure 10 is the robot motion path planned by the ant colony optimization algorithms. From the initial position to the target position, the robot’s motion path was very clear. Each solid circle in Figure 11 represents the position of the robot at each step on the map, while the straight lines in the circle indicate its upcoming posture. Compared with traditional standard maps, this non-standard map that combines real environments was simpler and clearer for robot motion path planning and pose planning. From the process of robot motion path planning under the above virtual non-standard map, it can be seen that, theoretically, we can obtain the robot’s planned path and pose by converting the non-standard real environment map into a standard grid map. Then, by returning the planned path and pose to a non-standard map, the robot’s motion path planning route and pose in the real environment map can be obtained. Next, we needed to analyze and study the entire path planning process through real non-standard environmental maps.

## 4. Experiment

### 4.1. Processing of Non-Standard Real Environment Map

Figure 12 show the calibration and non-standard real environment map preprocessing. The calibration object in the upper right corner of Figure 12a is very important as it ensures reference for later map size determination. Similarly, we conducted it in the Matlab environment. Figure 12b shows a non-standard map of the real environment that was read in. After reading Figure 12b, we boxed and cropped it (as shown in Figure 12c) to obtain the area for later standardized map design, as shown in Figure 12d. By binarizing Figure 12d, we can obtain Figure 12e,f. For the convenience of later processing, we chose Figure 12e as the object for grid processing.

### 4.2. Identification of Obstacles and Calibration Objects in Non-Standard Environmental Maps

After obtaining Figure 12e, we used the region props function to obtain the area parameters of obstacles and calibration objects, and we used the rectangle function for box selection, as shown in Figure 13. It is evident from Figure 13 that various obstacles were boxed, including the size calibration objects placed. We obtained a diameter of 183 for the calibration object in the system through box selection. As the box selection area parameter in the previous entire map was 960 × 960, the size of the entire image was 960 × 960. Because the actual size of the calibration object was 30 mm, based on the previous analysis, the scale obtained was 183/30 = 6.1. From this, we obtained the true environmental map size through data conversion, which was 960/6.1 = 157.377 mm.

### 4.3. Non-Standard Environment Map Generation

After binarization, we needed to segment the map to convert it into a grid map. We divided the map into 60 × 60 grids, which means a total of 3600 grids, and the size of the entire map was 960 × 960; therefore, the side length of each grid is
U = *a*/*n* = 960/60 = 16(10)

Next, we used the floor function to convert the matrix where the obstacle is located to 0, and the remaining areas to 1. Then, the fill function is used to fill the area with a median value of 0 in black, forming a standard grid map as shown in Figure 14. In order to facilitate the subsequent ant colony algorithms path planning, we also needed to code each grid. We divided the map into 60 × 60 parts, and the encoded map is shown in Figure 15.

Figure 15 real environment map grid code result is also very important. Through it, we can clearly understand the specific positioning of each part in the map.

### 4.4. Implementation of Ant Colony Optimization Algorithms

#### 4.4.1. Main Parameters

In this map, we set the starting position as point 80, the target position as point 3220, the number of iterations as 100, and the number of ants as 200. Based on the previous settings in Table 2, we obtained the parameters of the non-standard map for the actual environment, as shown in Table 4. After the parametric setting, we can carry out ant algorithms. 

#### 4.4.2. The Results and Analysis

Figure 16, Figure 17 and Figure 18 show the path planning map, pose planning map, and displacement iteration graph after the grid of non-standard map. They are the planned paths obtained through 100 iterations of ant colony algorithms. The path and poses planned by ant colony algorithm are clearly shown in the figure. At the same time, after 100 iterations, it can be seen from Figure 18 that the path planned by ant colony algorithms was generally regional optimization, and the final selected path was 274.233 mm. It should be noted that the path length here was obtained by converting it through a scale. The above results prove that the role of ant colony algorithm in path planning is very obvious. The planned path was optimized.

### 4.5. Integration of Planning Path with Non-Standard Real Environment Map

After planning the path and pose, we opened the originally selected non-standard map of the real environment. Then, we applied the parameters obtained through previous calculations to the graph. Figure 19 is the planning path, Figure 20 is the planning pose. From Figure 19, it can be seen that the planned path was clear and reasonable, avoiding obstacles and ultimately reaching the endpoint This also achieved the goal of this article. Compared to traditional standard maps, Figure 20 has a more guiding role in robot motion control.

### 4.6. The Impact of the Number of Iterations

Theoretically speaking, increasing the number of iterations can better find the optimal motion path of the robot, but the greater the number of iterations, the longer the operation time will be. On the one hand, in order to improve the effect of path planning of ant colony algorithm on actual environment map, we conducted a comparison of six sets of experiments with different iterations, as shown in Figure 21.

According to Figure 21, the path length and processing time changes of Figure 22 can be obtained. It can be seen from the figure that under the path planning of the existing ant colony algorithm in a real environment, when the number of iterations was 50 and 800, the moving path displacement was the minimum value of 169.3 mm, and the movement path displacement changed little between 50 and 800 times. However, the processing time of the algorithm increased significantly, and the operation time of 800 times was 13.98 times that of 50 times. Obviously, when other existing parameters remained unchanged, setting the iteration number 50 times can not only improve the operation efficiency but also obtain better-planned paths and poses. This also laid a good foundation for us to conduct other related experiments in the later stage. When the number of iterations was 50 times, the path planning diagram in the real environment was as shown in Figure 23.

### 4.7. Repeat Implementation

In order to ensure the universality of the method described in this article, we verified it through multiple experiments. We conducted the same experiment on six non-standard real environment maps in Figure 24 and obtained satisfactory planning paths. After multiple experimental verifications, the path planning method proposed in this article was true and effective in non-standard maps in real environments. The path planning method proposed in this article had broad application prospects.

## 5. Conclusions

With the development of technology, robots have become a common intelligent human-assistance tool. Because path planning is the core technology in robot motion control, its importance is obvious. In response to the problem that the maps used for robot path planning are often obtained through sensor measurement, which is time-consuming and laborious. This article studied the method of using of non-standard real environment map to plan the path. By placing calibration objects in the map, the generated non-standard planned paths can reflect the real environment, greatly improving the practicality of the path planning algorithm. In this paper, ant colony algorithms was used for path planning of non-standard map in a real environment, which greatly improved the intelligence of path planning algorithm. The experimental results of path planning based on ant colony algorithms for non-standard maps in virtual and real environments show that the method can realize path planning and pose planning of mobile robots in non-standard environments. Because the time of sensor measurement to obtain map was saved, the robot path planning was more efficient. Since the real environment map was used, the generated mobile robot path and pose were more intuitive. The research results of this article had great reference significance for improving the intelligence level of robot path planning methods in the later stage.

## Figures and Tables

**Figure 1 sensors-23-07502-f001:**
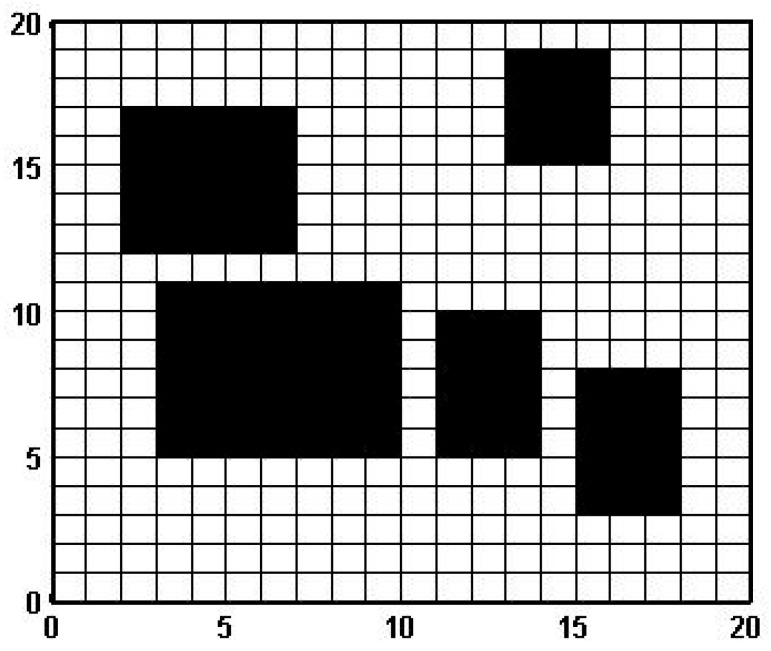
Common ant colony algorithm grid map.

**Figure 2 sensors-23-07502-f002:**
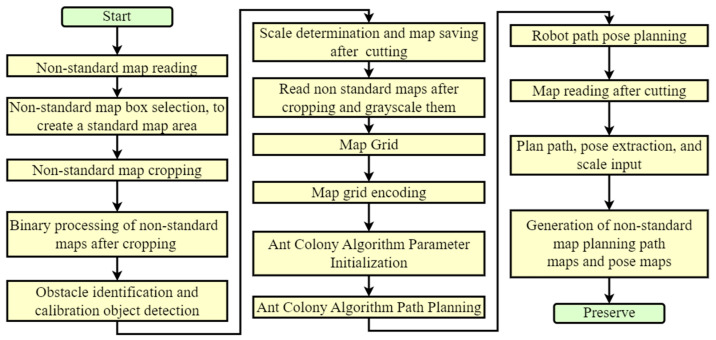
Non-standard map path planning approach based on ACO research method process.

**Figure 3 sensors-23-07502-f003:**
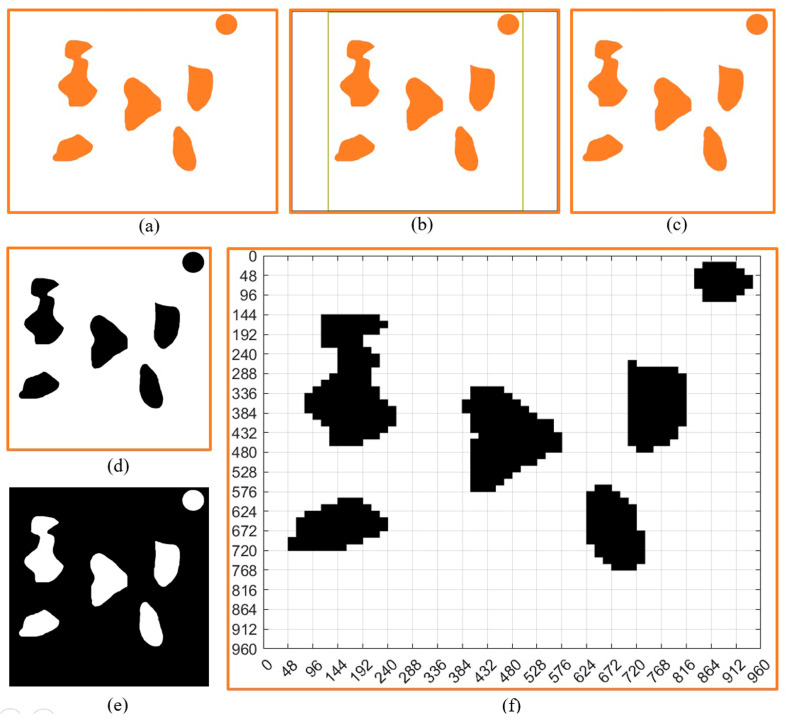
A diagram of the process of generating a grid map with a non-standard map. (**a**) Non-standard environment map. (**b**) Non-standard environment map box selection. (**c**) Non-standard environment map cutting. (**d**) Non-standard environmental map binarization result 1. (**e**) Non-standard environmental map binarization result 2. (**f**) Grid design for non-standard maps.

**Figure 4 sensors-23-07502-f004:**
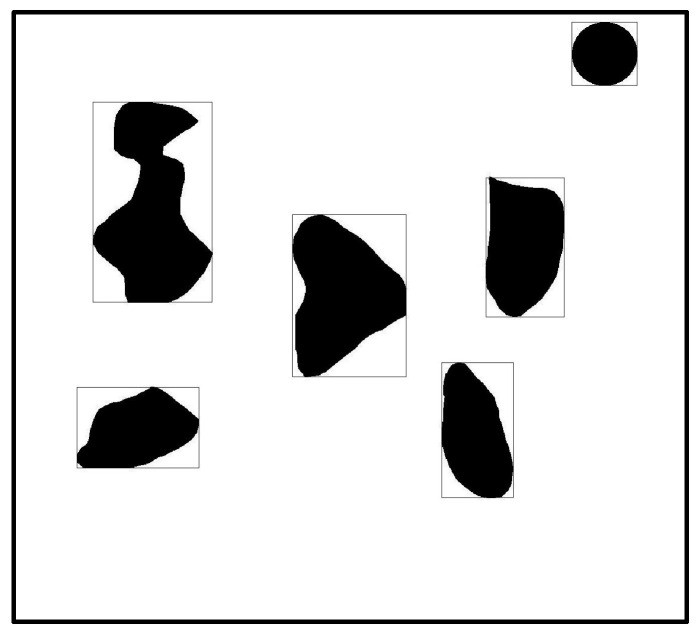
Obstacle identification and box selection.

**Figure 5 sensors-23-07502-f005:**
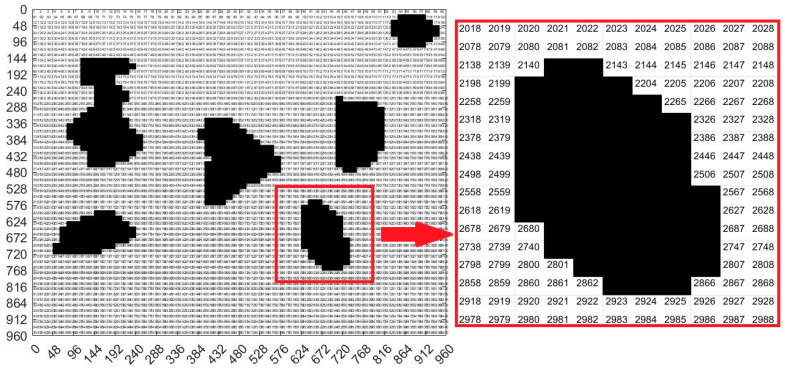
Display of standard map coding results.

**Figure 6 sensors-23-07502-f006:**
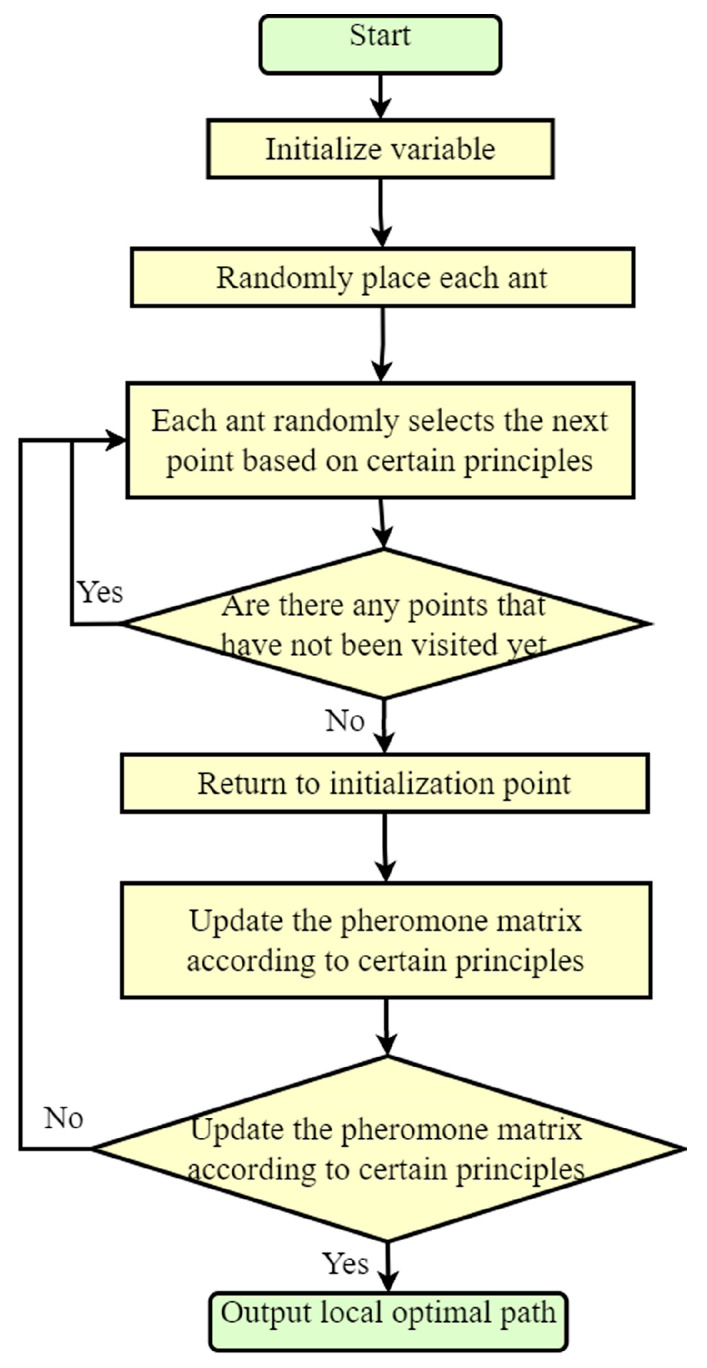
Principle of ant colony algorithms.

**Figure 7 sensors-23-07502-f007:**
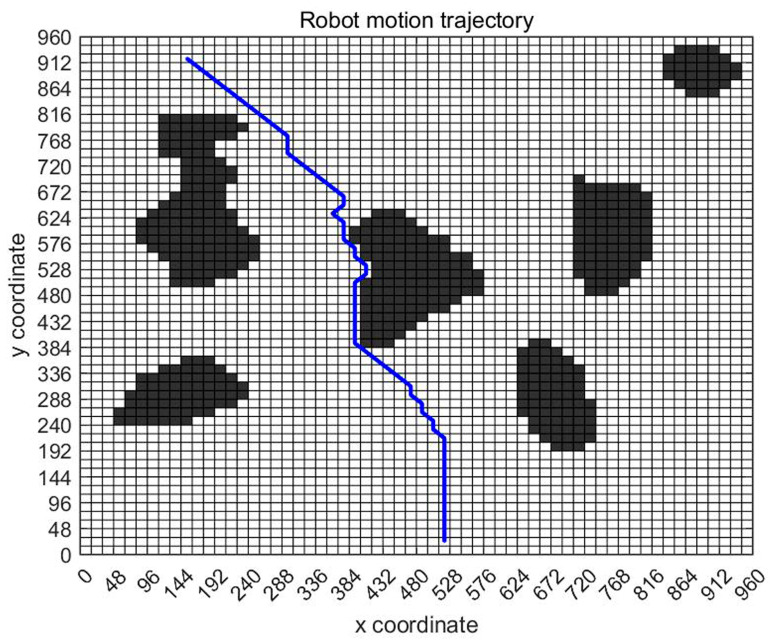
The generation of virtual map planning path.

**Figure 8 sensors-23-07502-f008:**
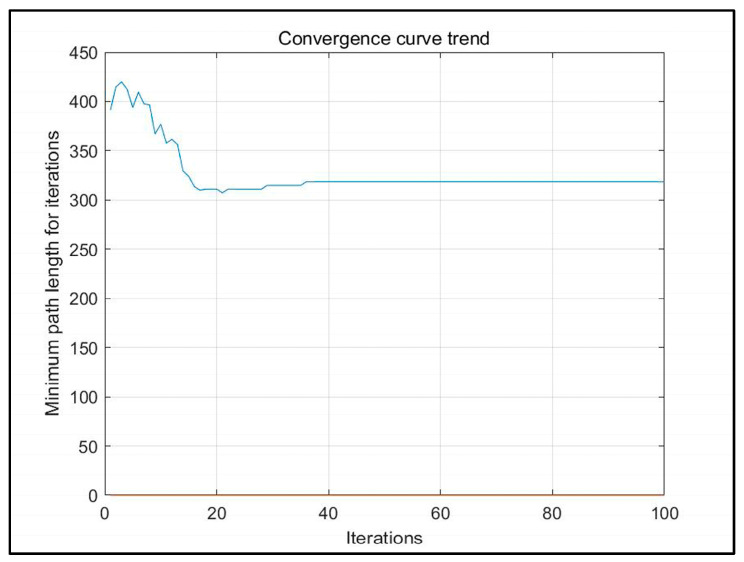
Displacement convergence curve chart of virtual map.

**Figure 9 sensors-23-07502-f009:**
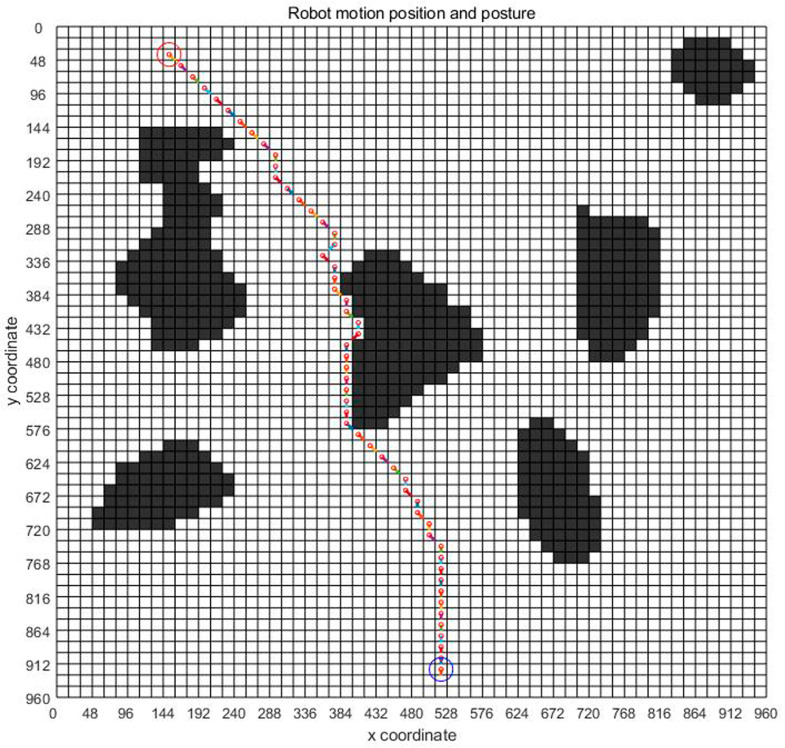
Pose planning of virtual map.

**Figure 10 sensors-23-07502-f010:**
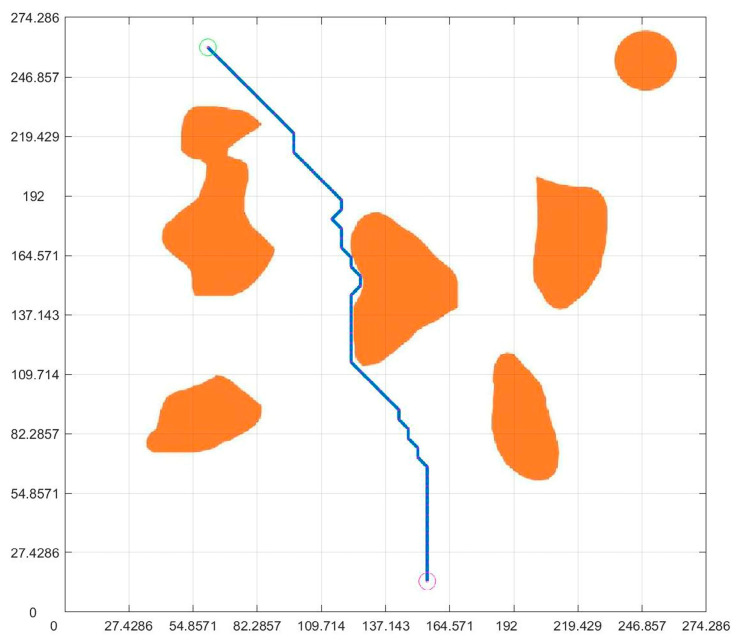
Virtual map and path fusion.

**Figure 11 sensors-23-07502-f011:**
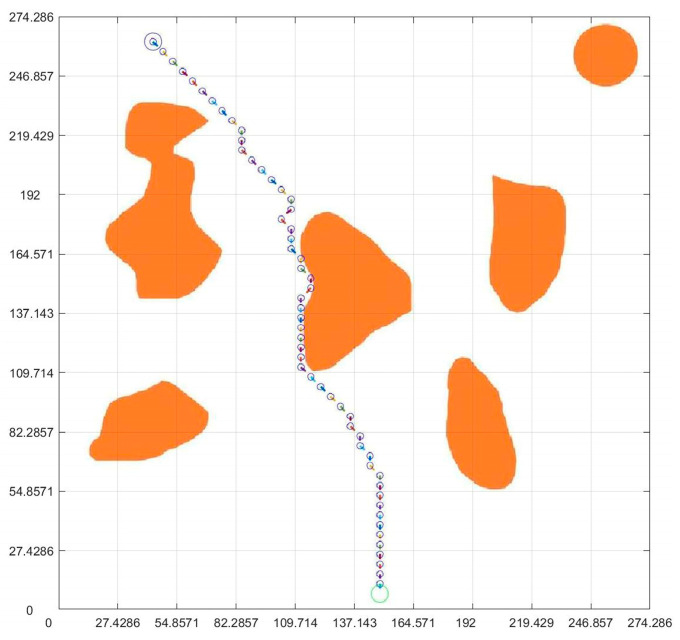
Virtual map and pose fusion.

**Figure 12 sensors-23-07502-f012:**
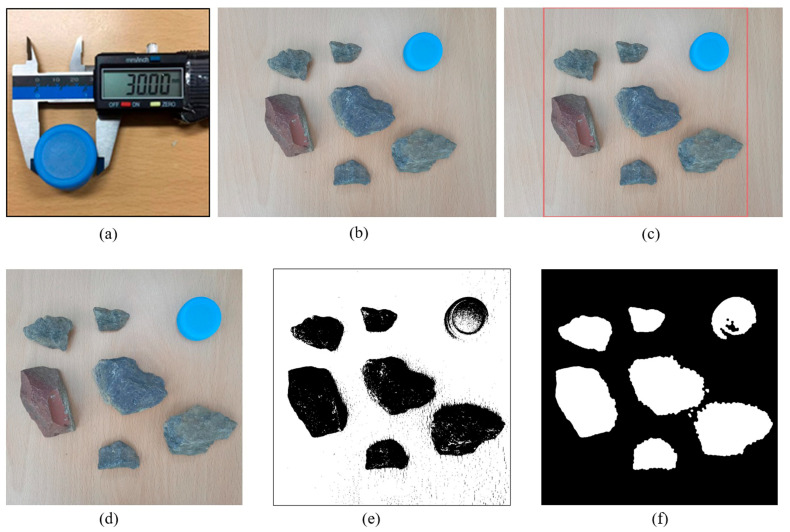
Calibration and non-standard real environment map preprocessing. (**a**) Size calibration object. (**b**) Non-standard real environment map. (**c**) Non-standard real environment map box selection. (**d**) Non-standard real environment map cutting. (**e**) Non-standard environmental map binarization result 1. (**f**) Non-standard environmental map binarization result 2.

**Figure 13 sensors-23-07502-f013:**
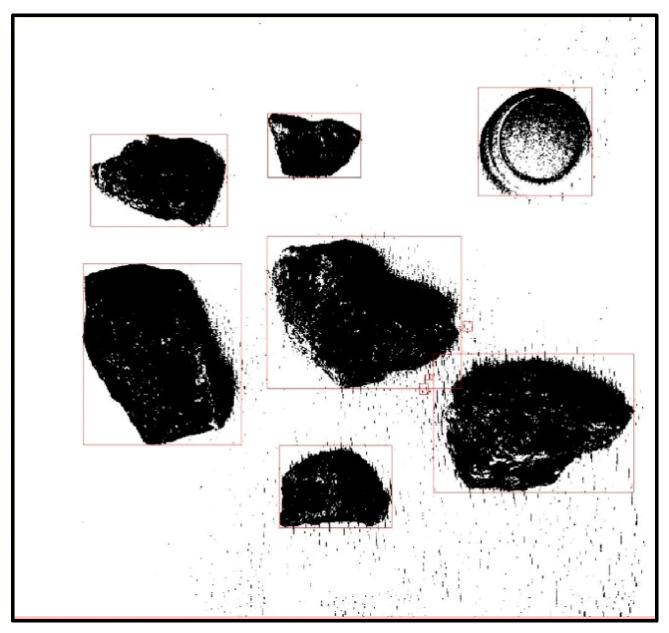
Identified obstacles in the real environment.

**Figure 14 sensors-23-07502-f014:**
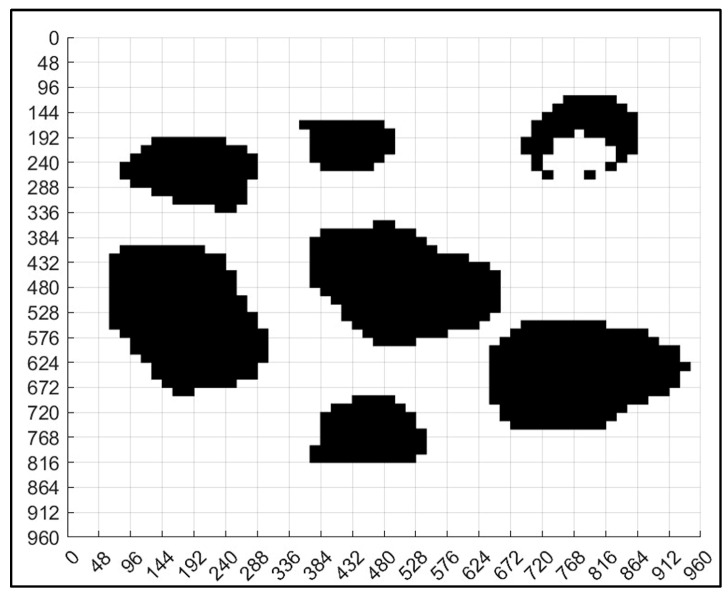
Non-standard real environment map grid design.

**Figure 15 sensors-23-07502-f015:**
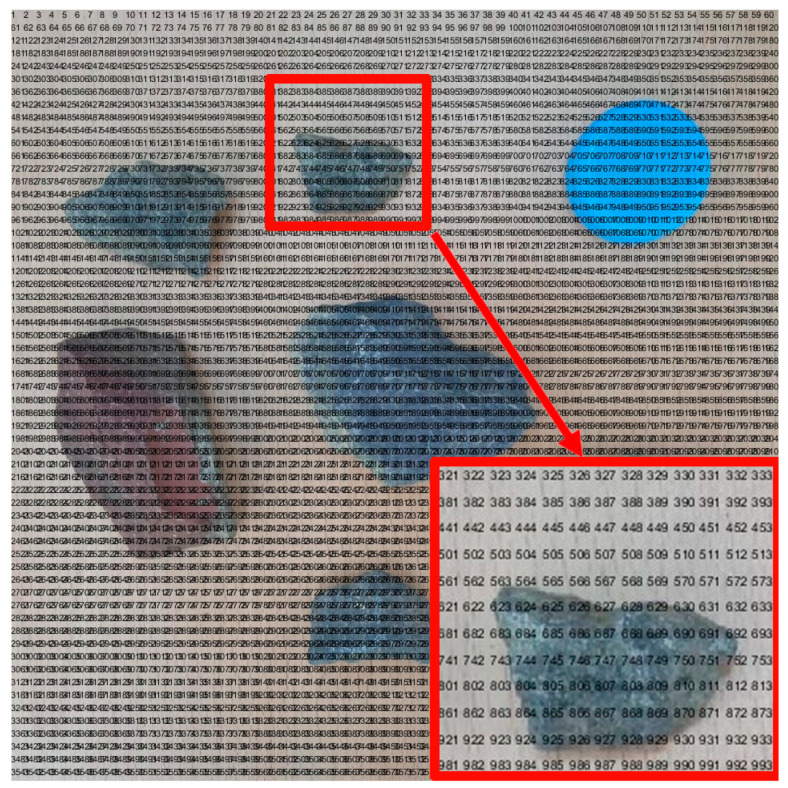
Real environment map grid code result.

**Figure 16 sensors-23-07502-f016:**
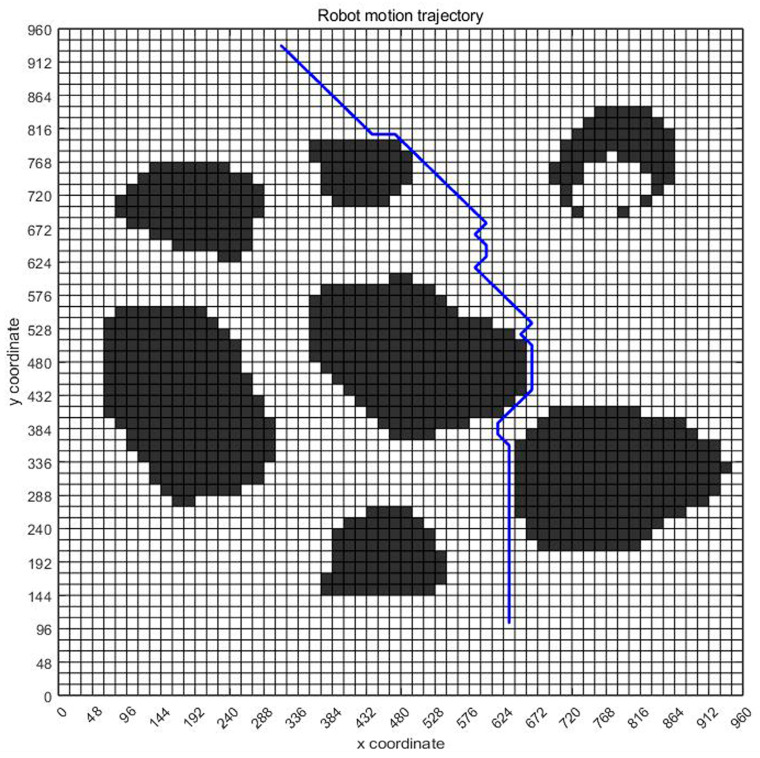
Standard map path planning result.

**Figure 17 sensors-23-07502-f017:**
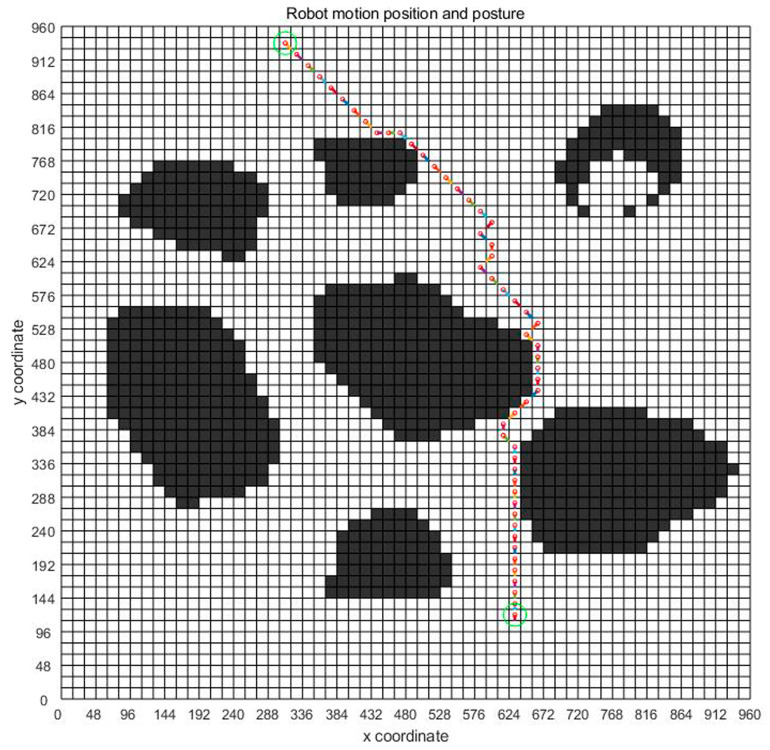
Standard map pose planning map.

**Figure 18 sensors-23-07502-f018:**
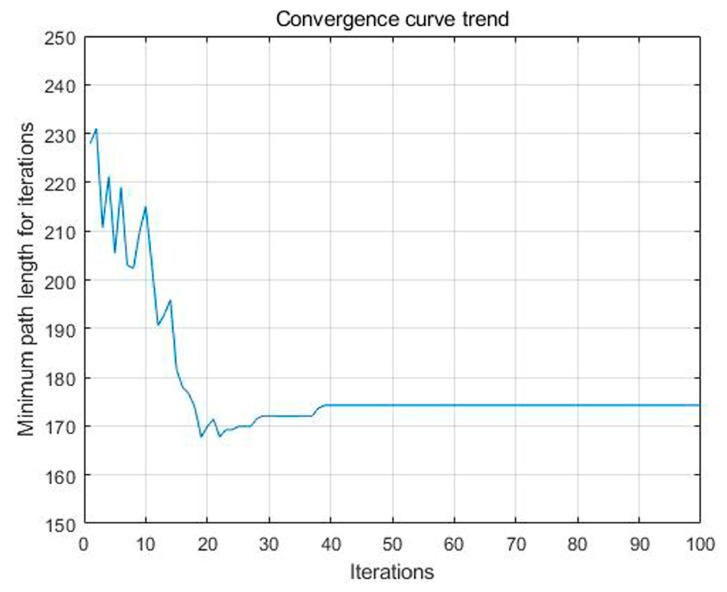
Displacement iteration graph of real environment.

**Figure 19 sensors-23-07502-f019:**
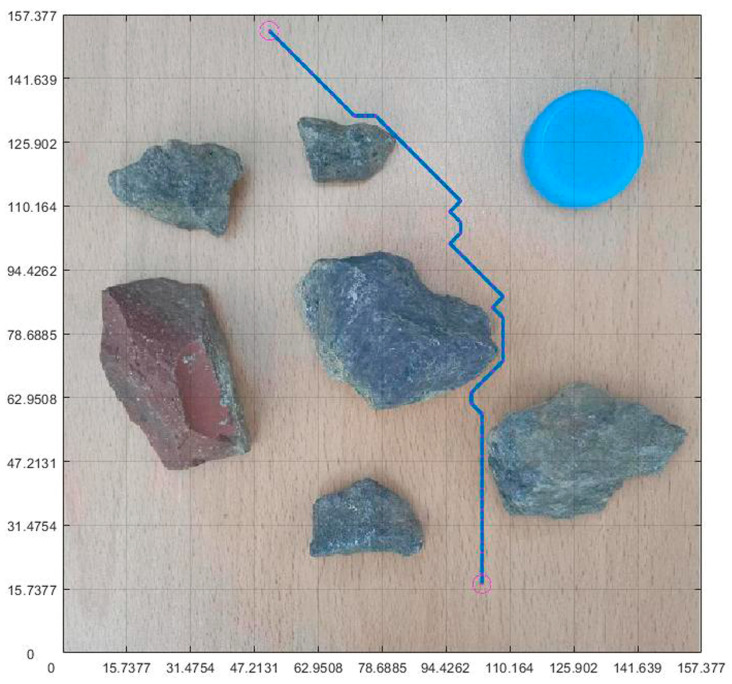
Integration of planning path and non-standard real environment map.

**Figure 20 sensors-23-07502-f020:**
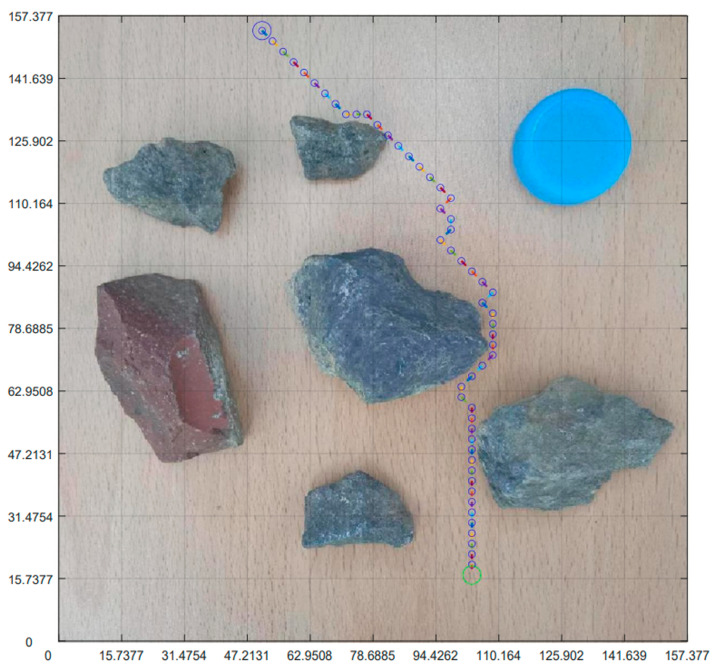
Integration of planning pose and non-standard real environment map.

**Figure 21 sensors-23-07502-f021:**
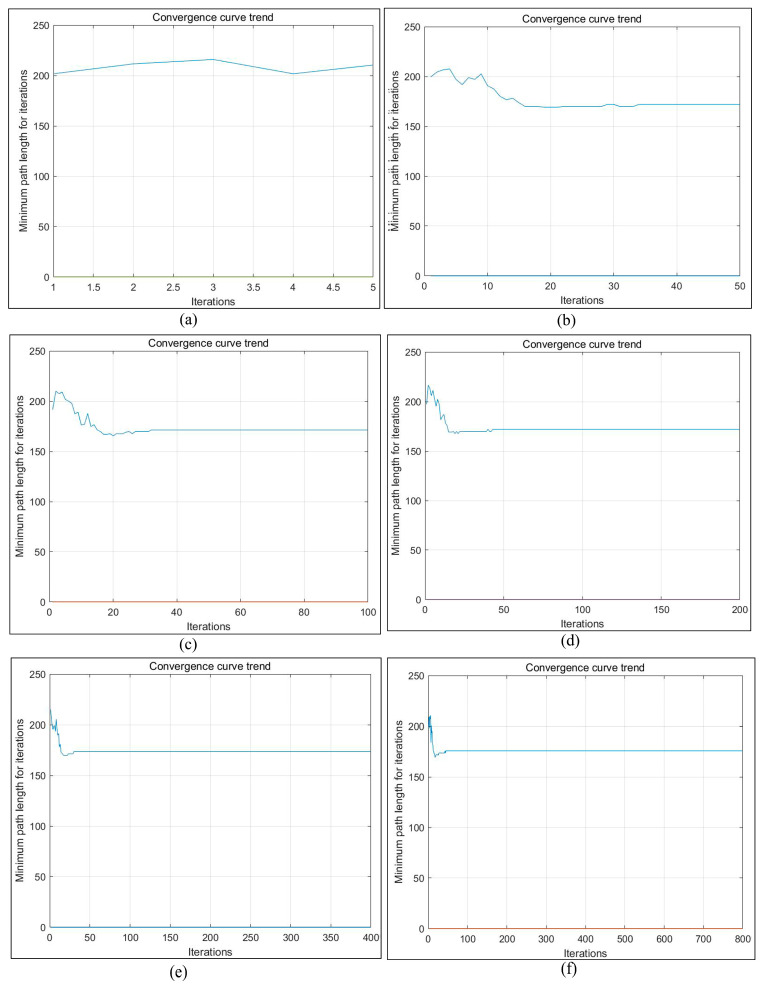
Convergence curves of different iterations under non-standard maps of real environment. (**a**) 5 iterations. (**b**) 50 iterations. (**c**) 100 iterations. (**d**) 200 iterations. (**e**) 400 iterations; (**f**) 800 iterations.

**Figure 22 sensors-23-07502-f022:**
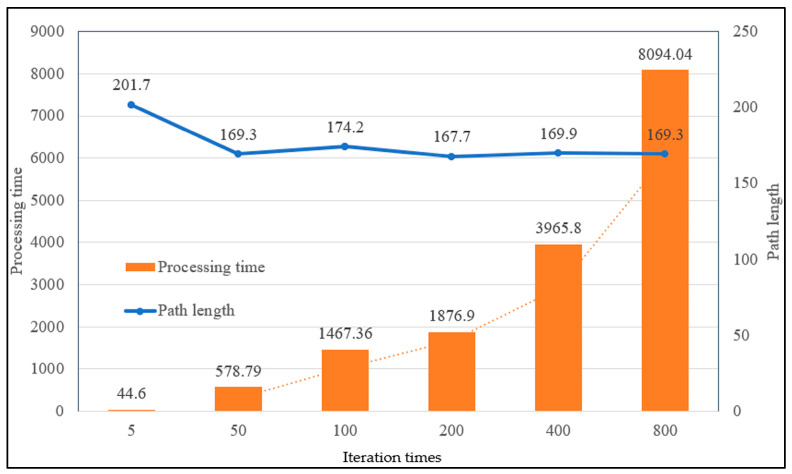
The changes in processing time and path length.

**Figure 23 sensors-23-07502-f023:**
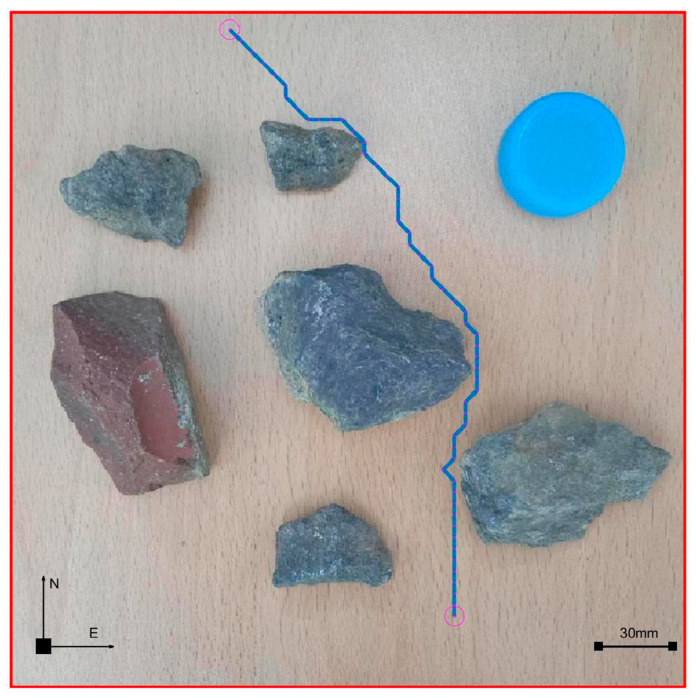
Integration of planning pose and non-standard real environment map with 50 iterations.

**Figure 24 sensors-23-07502-f024:**
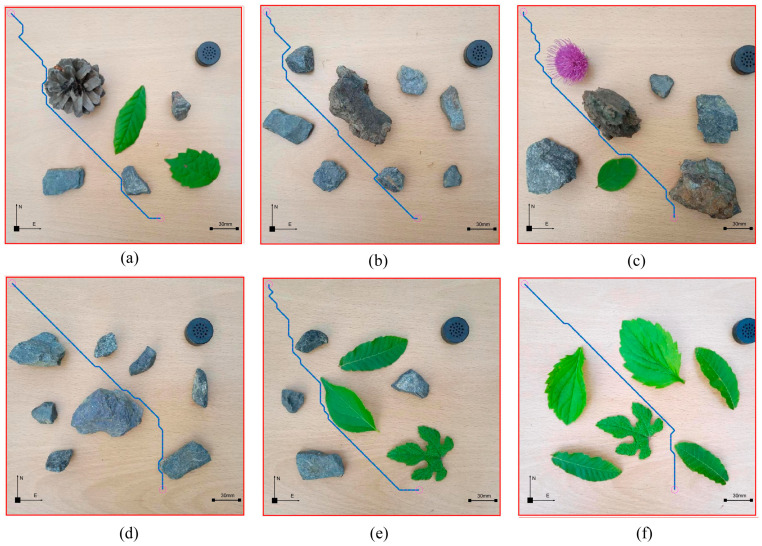
Planning paths under non-standard maps in different real environments. (**a**) Environment 1. (**b**) Environment 2. (**c**) Environment 3. (**d**) Environment 4. (**e**) Environment 5. (**f**) Environment 6.

**Table 1 sensors-23-07502-t001:** Comparison of different path planning methods using maps now.

Algorithm	Main Usage Maps	Algorithm Type	Real Environmental Map Used
ACO	Grid map, Octo Map	Search algorithm	No
A*	Grid map	Search algorithm	No
PRM	Grid map	Search algorithm	No
PRT	Grid map	Probabilistic path planning algorithm	No
RRT	Grid map, Octo Map	Probabilistic path planning algorithm	No
GA	Grid map,	Search algorithm	No
PSO	Grid map, Octo Map	Search algorithm	No

**Table 2 sensors-23-07502-t002:** Main parameters of virtual map for ant colony algorithm operation.

Parameter	*K*	*M*	*Alpha*	*Beta*	*Rho*	*Q*	*S*	*E*
Value	100	200	1.8	25	0.3	1	130	3513

**Table 3 sensors-23-07502-t003:** Initial and target position settings and passing points.

Initial Position: 130	Target Position: 3513
130 191 252 313 374 435 496 557 618 679 739 799 860 921 982 1043 1104 1164 1223 1284 1344 1404 1465 1525 1586 1646 1705 1765 1825 1885 1945 2005 2065 2125 2186 2247 2308 2369 2430 2490 2551 2611 2672 2732 2793 2853 2913 2973 3033 3093 3153 3213 3273 3333 3393 3453 3513	

**Table 4 sensors-23-07502-t004:** Parameters of ant colony algorithms.

Parameter	*K*	*M*	*Alpha*	*Beta*	*Rho*	*Q*	*S*	*E*
Value	100	200	2	30	0.3	1	80	3220

## Data Availability

Data available on request due to restrictions, e.g., privacy or ethical.

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
