# Peer review of "Non-Standard Map Robot Path Planning Approach Based on Ant Colony Algorithms"

_sensors, 2023, doi:10.3390/s23177502_

Round 1

Reviewer 1 Report

The manuscript is about the path planning problem of solving mobile robots with ACO algorithm, which is an interesting topic. Here are some comments:

1. Is the core work of the manuscript the conversion of photographs to grids, or is it optimal path planning for mobile robots?

2. Use the standard paradigm of optimization theory to redefine the objective function and constraints of the optimization model, i.e., the optimal path.

3. From the convergence graph, the result of ACO does not converge.

4. From the results, the chosen path is not the optimal path.

5. Improve the ACO optimization algorithm to obtain better optimization results and compare it with native ACO, Dijkstra's algorithm, A * algorithm, etc.

6. If there are experimental conditions, please use the simulation results to guide the mobile robot to complete the path walking.c

Reviewer 2 Report

Nowadays, robots have become indispensable helpers of humans in various areas of life. Planning the path of the robot's movement is therefore a very important task that allows the robot to perform its function efficiently. The article focuses on alternative solutions for robot path planning. The authors propose a robot path planning method based on ant colony algorithms after a standardized design of custom map grids.

In my opinion, the topic is interesting and quite innovative.

Using ant colony algorithms to plan the path of a custom map in a real environment is a good idea to solve the presented problem as it can improve the intelligence of the path planning algorithm. This article is worded correctly and has good consistency logic. The article presents a series of examples with illustrations and correctly described formulas to confirm the correctness of the presented method and to enable the reader to understand the method used.

However, I have comments to the authors:

1. on page 4 row 151 the word "Methods" is probably not needed,

2. page number 4 template (2), the "w" template and the translation should use the same font,

3. on page 5, the subsection is "2.3.3 Post-processing of path planning algorithms for ant colonies" and "2.3.3. Related works in this article"

4. page No. 7, figures 3, 4, 5, 6, 7, 8 are not well marked, each drawing should be marked with its number.

5. page number 9, figure 10 is unclearly presented, I propose a correction,

6. page No. 13, same as comment No. 4,

7. page no. 14, same as comment no. 4.

Reviewer 3 Report

sensors-2526229

This article presents a Non-standard Map Robot Path Planning Technology Based on Ant Colony Algorithm. The topic is timely new and interesting but the work presented here possesses several flaws, such as:

1.     The title seems strange. I suggest to add a title as “Non-standard Map Robot Path Planning Approach Based on Ant Colony Algorithm

2.     Some numerical terms seem incorrect mathematically, such as N={…..} and after equation (2) w=….. and so on.

3.     There are numerous typos and grammatical errors that must be corrected with careful revision of the overall manuscript.

4.     Add a full stop at the end of every caption.

5.     Section 2.3.2 can be “Implementation” and section 2.3.3 can be “Post-processing” because this section already belongs to ant colony, so do not need to repeat. Moreover, section 2.3.3. can be “Related work”

6.     The key contribution of this work should be added in bullet form at the last second paragraph of the introduction section.

7.     First paragraph of section 3. Methods, must be split into 2 brief and concise paragraphs. The current is too lengthy and wordy.

8.     The authors have added some captions but there is no figures such as “Figure 4. Non-standard environment map box selection”,  “Figure 5. Non-standard environment map cutting”, “Figure 7. Non-standard environmental map binarization result 2”, “Figure 8. Obstacle identification and box selection.” Similarly, check figures 19, 20, and so on. The authors should add a single caption and name the three sub-figures with part (a), (b), and (c). Moreover, briefly explain each part in the caption. The authors should look at some standard articles to understand how to organize an article.

9.     Most the tables and figures captions are monotonous, this must be avoided.

10.  The term C = {(x1, y1), (x2, y2) ..., (xi, yi), ..., (xn, yn)}}; just before equation (7) is incorrect. All numerical equations and terms should be revised carefully.

11.  The results illustrated in figure 13 are useless.

12.  Most of the figures have very low resolution. All of them must be modified and add high resolution figures. Additionally, the text in some figures are unable to read.

13.  The technical depth of the paper is not adequate.

14.  The flow is completely missing. Also, avoid the unnecessary figures so not to tends to confusion.

15.  The authors have stated that “Compared to traditional standard maps, this type of map has a more guiding role in robot motion control” but in fact there is no proper comparison with other state-of-the-art approaches related to robot motion control. I strictly recommend to properly compare their model with other state of the art approaches and add a separate table or graph for better understanding.

16.  According to the paper length and work presented, the current references are very limited. Further references should be added, especially from the last 3-5 years.

17.  At last, the English writing of this article is very low standard, it must be improved.

Extensive editing of English language required

Round 2

Reviewer 1 Report

The authors have addressed my concerns and I believe the paper is now suitable for acceptance.

Author Response

Thank you very much!!

Reviewer 3 Report

NA

Minor editing of English language required

Author Response

Thank you very much!!
